# Host Serum Amyloid A1 Facilitates *Streptococcus pneumoniae* Adaptation to Acidic Stress Induced by Pneumococcal Anaerobic Metabolism

**DOI:** 10.3390/microorganisms13061309

**Published:** 2025-06-04

**Authors:** Weichen Gong, Masayuki Ono, Tomoko Sumitomo, Momoko Kobayashi, Yujiro Hirose, Shigetada Kawabata

**Affiliations:** 1Department of Animal Food Function, Graduate School of Agricultural Science, Tohoku University, Sendai 980-8577, Japan; 2Department of Microbiology, Graduate School of Dentistry, Osaka University, Osaka 565-0871, Japanu914792j@ecs.osaka-u.ac.jp (M.K.); kawabata.shigetada.dent@osaka-u.ac.jp (S.K.); 3Department of Oral Microbiology, Graduate School of Biomedical Sciences, Tokushima University, Tokushima 770-0855, Japan; 4Center for Infectious Diseases Education and Research, Osaka University, Osaka 565-0871, Japan

**Keywords:** serum amyloid A1, *Streptococcus pneumoniae*, formic acid

## Abstract

*Streptococcus pneumoniae* (*S. pneumoniae*) is a leading cause of pneumonia, and its interaction with host acute-phase proteins remains underexplored. Serum amyloid A1 (SAA1), an acute-phase protein, plays a crucial role in immune modulation. This study investigates the role of SAA1 in the early stages of respiratory infection by *S. pneumoniae* and its potential contribution to bacterial adaptation under acidic stress. We used a murine nasal infection model to simulate the early phase of *S. pneumoniae* invasion into the lower respiratory tract. Levels of SAA1 and C-reactive protein (CRP) in bronchoalveolar lavage fluid (BALF) and serum were quantified using ELISA. In vitro assays examined the effect of serum and recombinant SAA1 on bacterial survival under acidic conditions. Fluorescence-labeled recombinant SAA1 and microscopy were utilized to assess SAA1 internalized by *S. pneumoniae*. Following nasal infection, SAA1 levels in BALF were significantly reduced, whereas CRP levels remained unchanged. In vitro, serum enhanced *S. pneumoniae*’s resistance to acidic byproducts including formic, lactic, and acetic acids. Specifically, formic acid promoted bacterial uptake of SAA1, and this internalization improved bacterial tolerance to acidic conditions. Fluorescence microscopy confirmed that SAA1 is internalized by *S. pneumoniae*. *S. pneumoniae* can internalize SAA1 to bolster resistance to acid stress, particularly formic acid. This study reveals a novel host–pathogen interaction mechanism wherein *S. pneumoniae* exploits host acute-phase proteins for environmental adaptation, offering new insights into bacterial survival strategies during infection.

## 1. Introduction

Streptococcus *pneumoniae* (*S. pneumoniae*) is a common commensal bacterium in the oropharynx, primarily transmitted via aerosolized droplets. Upon entering the body through the nasal or oral routes, it colonizes the nasopharyngeal or oral mucosa. If it escapes the protective barriers of the upper respiratory tract, such as mucus, cilia, and secretory antibodies, or if the immune system is weakened due to factors such as viral co-infection, aging, or chronic diseases, the bacteria can reach the lungs. In the lungs, they attach to alveolar epithelial cells, avoid being cleared by immune cells, and trigger inflammation, which leads to pneumonia [1,2].

Neutrophils are the first and most abundantly recruited leukocytes during *S. pneumoniae* infection. Impaired neutrophil function is associated with increased susceptibility in infected mice, underscoring their essential role in host defense [3]. Neutrophils eliminate bacteria via phagocytosis, production of reactive oxygen species (ROS), and secretion of antimicrobial peptides. Although neutrophil plays a crucial role in *S. pneumoniae* acute infection phase, the overactivation of neutrophils causes an accumulation of fluid in the alveoli, with excessive neutrophils even causing edematous and hemorrhagic lungs [4,5,6,7]. The inflammatory response caused by neutrophils can also increase permeability of the lung capillaries so that more serum proteins and immune cells infiltrate the alveolar cavity, exacerbating gas exchange disruption and creating an anaerobic environment [8,9].

As a facultative anaerobic organism, *S. pneumoniae* lacks the cytochromes and heme-containing proteins involved in aerobic respiration. Although *S. pneumoniae* lacks mechanisms for coping with oxidative stress, it is aerotolerant and showed capacity to survive under atmospheric oxygen pressure [10]. These strategies include the production of superoxide dismutase (SodA), which converts superoxide radicals into hydrogen peroxide, and thiol peroxidase (TpxD), which helps reduce peroxides [11,12,13]. In addition, NADH oxidase and other flavoproteins help manage oxidative stress by maintaining redox balance within the cell. The first step of pneumococcal aerobic pathway is to convert pyruvate to acetyl phosphate, which is catalyzed by pyruvate oxidase (SpxB). SpxB catalyzes a two-electron reduction in O_2_, thereby forming the potentially damaging compound H_2_O_2_ [14]. In addition, *S. pneumoniae* is known to maintain a fermentative metabolism in anaerobic environments; it converts pyruvate to lactate via the lactate dehydrogenase (LDH) [15]. Besides lactate, most of the pyruvate can be converted to acetyl-CoA and formate by the oxygen-sensitive pyruvate formate lyase (PFL) [16]. Pneumococcal acid exhibits strong survival capacity under acidic stress conditions, such as the acidic environment of inflammatory foci, and tolerates lethal pH through a mechanism known as the acid tolerance response. Pneumococcal acid’s tolerance response mechanisms include the upregulation of F0F1-ATPase, which helps maintain intracellular pH by pumping protons out of the cell, and the activation of chaperone proteins like GroEL and DnaK, which prevent acid-induced protein denaturation. Additionally, arginine deiminase pathway (ADI) enzymes help generate ammonia, which buffers cytoplasmic acidity [17,18,19]. *S. pneumoniae* activates acidic stress-induced lysis in response to acidic environments, favoring the release of cell wall compounds, DNA, and virulence factors [20]. This process is primarily mediated by the autolysin LytA, which is tightly regulated and can be triggered under acidic pH. The released DNA also contributes to extracellular matrix stability in biofilms, further protecting surviving bacteria from immune attack and antibiotics [21,22,23].

During infection, acute-phase proteins such as CRP and SAA1 leak into the alveolar space due to increased pulmonary vascular permeability. Stimulated by IL-6, CRP facilitates complement activation and enhances phagocytosis [24]. Similarly, IL-6 induces rapid SAA1 synthesis, which is a more sensitive early marker of bacterial infection than CRP, often increasing over 1000-fold [25,26]. SAA1 regulates the chemotaxis of inflammatory cells, attracting neutrophils and monocytes to the infection site, and interacts with TLR2/4 to activate immune cells [15]. Previous studies also indicate that CRP and SAA1 contribute to immune defense against bacteria through mechanisms such as direct bacterial binding, complement system activation, and enhanced phagocytosis [27,28]. For example, CRP binds to phosphocholine on the C-polysaccharide of *S. pneumoniae*, activating complement via the classical pathway and enhancing phagocytosis [29]. Likewise, SAA1 binds to the outer membrane protein A (OmpA) of Gram-negative bacteria such as *Escherichia coli* and *Pseudomonas aeruginosa*, boosting neutrophil-mediated phagocytosis [30]. Moreover, SAA1 exhibits bactericidal activity against *Staphylococcus aureus* under acidic conditions, likely through membrane disruption [31].

Despite the recognized roles of CRP and SAA1 in modulating immune responses during infection, the specific function of SAA1 in the early stage of *S. pneumoniae* infection remains poorly defined [32,33]. In this study, we aimed to elucidate whether host-derived SAA1 contributes to host protection or instead facilitates bacterial adaptation during the initial 24 h of pneumococcal infection, particularly under inflammatory and acidic stress conditions.

## 2. Materials and Methods

Our research complied with all of the relevant ethical regulations. All of the animal procedures were conducted according to the protocols approved by Animal Care and Use Committee of Osaka University Graduate School of Dentistry (authorization number 04-018-0).

All methods are reported in accordance with the ARRIVE guidelines (Animal Research: Reporting of In Vivo Experiments) to ensure rigorous and transparent reporting of animal research (https://arriveguidelines.org, accessed on 20 January 2025).

### 2.1. Bacterial Culture

*S. pneumoniae* D39 (serotype 2 clinical isolate) and TIGR4 (virulent serotype 4 clinical isolate) strains, both purchased from commercial suppliers, were cultured in Todd–Hewitt broth (Becton, Dickinson and Company [BD], Franklin Lakes, NJ, USA) supplemented with 0.2% yeast extract (BD) (THY broth) at 37 °C in 5% CO_2_ incubator.

### 2.2. Animal Infection Experiments

Female Slc:ICR mice at 6 to 7 weeks old (SLC Japan, Inc., Shizuoka, Japan) were used as the animal infection model in this study. The *S. pneumoniae* D39 strain was grown to the early exponential phase (The culture was initiated at an OD_600_ of 0.003 and incubated at 37 °C for 4 h, reaching an OD_600_ of 0.1.) and then washed with and resuspended in PBS.

For intranasal infection experiments, each mouse was intraperitoneally given 250 µL of anesthetic consisting of 20 µL midazolam (Takeda Pharmaceuticals, Osaka, Japan), 7.5 µL domitor (ZENOAQ, Fukushima, Japan), 25 µL vetorphale (Meiji Animal Health Co., Ltd., Kumamoto, Japan), and 197.5 µL PBS solution, then bacteria were given to mice by administration of 4 × 10^6^ CFU in 20 µL PBS. Blood and BALF were collected from mice 24 h post-intranasal infection after mice were euthanized by intraperitoneal injection of 300 µL pentobarbital sodium solution (20 mg/mL, Product mumber: P0776, Tokyo Chemical Industry Co., Ltd., Tokyo, Japan). BALF were collected by slowly drawing the injected PBS (1 mL) back into the syringe and serum was obtained from blood by centrifugation (8500 rpm, 15 min, 4 °C). Blood was collected by performing cardiac puncture immediately after euthanizing the mice. This method typically yields approximately 600–700 µL of blood per mouse. The collected BALF was defined as the original (undiluted) concentration, and serial dilutions were subsequently performed to measure SAA1 levels.

### 2.3. Quantification of SAA1 and CRP Using ELISA

After the collection of BALF from mice, SAA1 and CRP quantification was performed by ELISA following the manufacturer’s protocols described in the mouse serum amyloid A ELISA Kit (SAA; Cat# KMA0021, Invitrogen, Thermo Fisher Scientific, Waltham, MA, USA) and mouse CRP ELISA Kit (Cat# EM20RB, Invitrogen, Thermo Fisher Scientific, Waltham, MA, USA).

### 2.4. Incubation of S. pneumoniae with THY Broth Containing 20% Serum In Vitro

Serum was collected from Slc:ICR mice at 6 to 7 weeks old (SLC Japan, Inc., Shizuoka, Japan) and the SAA1 concentration were measured by the mouse serum amyloid A ELISA Kit (SAA; Cat# KMA0021, Invitrogen, Thermo Fisher Scientific, Waltham, MA, USA). *S. pneumoniae* D39 strain or TIGR4 strain was grown to the early exponential phase (OD_600_ of 0.1). Then, 5 µL of bacterial culture were added into 20 µL serum and 175 µL THY broth (containing final 0.05% formic acid, *v*/*v*) in a 96-well plate and incubated in 5% CO_2_ incubator for 12 h. To test whether SAA1 reduction was protease-dependent, a protease inhibitor (cOmplete™, EDTA-free Protease Inhibitor Cocktail, Roche, Basel, Switzerland) was dissolved and used following its instruction for use. Protease inhibitors were used at 1×, 2.5×, and 5× working concentrations to investigate whether the degradation of SAA1 by *S. pneumoniae* is dependent on protease activity.

After incubation, the supernatant of bacterial culture was collected after centrifugation (12,000 rpm, 10 min, 4 °C) and the remaining SAA was measured using the mouse serum amyloid A ELISA Kit (SAA; Cat# KMA0021, Invitrogen, Thermo Fisher Scientific, Waltham, MA, USA).

### 2.5. AF488-Recombinant SAA1 Intake by S. pneumoniae Using Fluorescence Microscope

Recombinant SAA1(rSAA1) (Cat# 2948-SA-025, R&D Systems, Minneapolis, MN, USA) was conjugated with AF488 following the protocol described in Alexa Fluor^®^ 488 Conjugation Kit (Fast)—Lightning-Link^®^ (Abcam, Cat# ab236553, Abcam Limited, Cambridge, UK). *S. pneumoniae* TIGR4 strain was grown to the early exponential phase (OD_600_ of 0.1) and 1 mL of bacteria were collected and resuspended with 50 µL PBS containing AF488-rSAA1 (final concentration = 0.25 µg/mL) in the presence of 0.05% formic acid (a non-toxic concentration) or 0.25% formic acid (a lethal concentration). Then, bacteria were incubated in 5% CO_2_ incubator, 5 µL of bacteria were collected after 1 h, 2 h, and 3 h, and observed using fluorescence microscope (KEYENCE BZ-X810, Osaka, Japan). Bacterial pellets were collected at 1 h, 2 h, and 3 h post-incubation through centrifugation (12,000 rpm, 10 min, 4 °C) and then added with 0.05% Triton X-100 to permeabilize bacteria, with their fluorescence intensity (Ex/Em = 485/535) measured after permeabilization for 1 h.

In the experiment showing recombinant SAA1 to enhance *S. pneumoniae* resistance against formic acid, we cultured *S. pneumoniae* in 0.25% formic acid together with AF488-rSAA1 (final concentration = 0.25 µg/mL). After 2 h incubation, Propidium Iodide was added to bacteria (final concentration = 1 µg/mL) and incubated at 37 °C for 15 min. After incubation, bacteria were washed twice and resuspended with PBS. Then, the bacteria were observed under a fluorescence microscope (KEYENCE BZ-X810, Osaka, Japan).

### 2.6. pH Measurement of Bacterial Culture Under Aerobic and Anaerobic Environments

A volume of 500 µL of *Streptococcus pneumoniae* D39 (OD_600_ = 0.003) was added in each well of a 24-well plate, then cultured in aerobic (5% CO_2_ incubator) and anaerobic environments created by using AnaeroPack-Anaero (MITSUBISHI GAS CHEMICAL Co., Inc., Tokyo, Japan). The supernatant of bacterial cultures was collected at different time points (1, 3, 5, 7, 9 h). A volume of 50 µL of 0.04% (*w*/*v*) phenol red solution (prepared in double-distilled water, Cat# 165-01121, FUJIFILM Wako Pure Chemical Corporation, Osaka, Japan) was mixed with 950 µL bacterial supernatant collected at each point and the absorbance at 550 nm was measured. A calibration curve was determined in phenol red broth adjusted to pH values ranging from 4 to 10. For THY broth, the culture supernatant at each point was supplemented with 0.04% phenol red and the absorbance was measured at 550 nm. A calibration curve was determined in THY broth, which was supplemented with 5 mg/mL phenol red and adjusted to pH values ranging from 3 to 10 (Appendix A, Appendix A).

### 2.7. Bacterial Growth Curve in Lethal Formic, Lactic and Acetic Acid Concentration

*S. pneumoniae* D39 strain was grown to the early exponential phase (OD_600_ of 0.1). Then, 5 µL of bacterial culture were added into 40 µL serum and 155 µL THY broth (containing 0.125% acetic, 0.1875% lactic, and 0.075% formic acid, *v*/*v*) in a 96-well plate and incubated in 5% CO_2_ incubator for 16 h. The OD_600_ of each well was recorded by a microplate reader (Infinite^®^ M Plex, Tecan, Männedorf, Switzerland).

### 2.8. Statistical Analysis

All results were statistically analyzed using the Student’s *t*-test in Excel (v. 16.81; Microsoft, Redmond, WA, USA). Each experiment was independently repeated at least three times, and data were collected from three biological replicates per group unless otherwise stated. Differences were considered statistically significant at *p* < 0.05.

### 2.9. Data Visualization

Data visualization was performed using Hiplot (ORG) (https://hiplot.org, accessed on 2 September 2024), a comprehensive and user-friendly web service designed for generating publication-quality biomedical graphics.

The other data visualization was initially created using Microsoft Excel (v. 16.81; Microsoft, Redmond, WA, USA) and subsequently refined with Keynote (v. 14.1; Apple Inc., Cupertino, CA, USA).

## 3. Results

### 3.1. SAA1 in BALF Significantly Decreases, While CRP Levels Remain Unchanged Following Nasal Infection with S. pneumoniae

To mimic the natural respiratory infection route of *S. pneumoniae*, we established a nasal infection model in mice. Mice were inoculated intranasally with 4 × 10^6^ CFU of *S. pneumoniae* in 20 µL PBS and analyzed 24 h post-infection. Bacteria were detected in BALF but not in blood (Figure 1A), confirming successful colonization of the lower respiratory tract without systemic spread. Given that acute-phase proteins typically increase during early bacterial infections, we measured CRP and SAA1 levels in BALF by ELISA. Surprisingly, CRP levels remained unchanged after infection (Figure 1B), while SAA1 levels significantly decreased (Figure 1C).

These findings suggest that CRP and SAA1 in BALF may not serve as reliable biomarkers for early-stage *S. pneumoniae* infection. Moreover, the decline in SAA1 post-infection implies that *S. pneumoniae* actively reduce SAA1 levels, possibly via proteolysis or endocytosis.

### 3.2. Serum Components Enhance S. pneumoniae Survival Under Acidic Stress Caused by Anaerobic Metabolism

*S. pneumoniae* infection disrupts the pulmonary epithelial barrier, increasing alveolar capillary permeability, allowing serum components to enter the alveoli. This infiltration elevates alveolar fluid volume, impairs gas exchange, and fosters a low-oxygen environment, which in turn promotes the bacterium’s fermentative metabolism [34]. Under such conditions, *S. pneumoniae*’s ability to produce acidic substances is enhanced (Figure 2A). KEGG analysis confirmed that under anaerobic conditions, *S. pneumoniae* produces elevated levels of formic, lactic, and acetic acids (Figure 2B). To simulate the presence of serum components in alveolar fluid, we supplemented THY medium with 20% mouse serum (*v*/*v*). This addition significantly enhanced *S. pneumoniae* survival in the presence of lethal concentrations of formic acid (0.075%, *v*/*v*), lactic acid (0.1875%, *v*/*v*), and acetic acid (0.125%, *v*/*v*) (Figure 2C–E).

These results indicate that serum components present in the inflamed alveoli may promote *S. pneumoniae* adaptation to acidic conditions caused by its own anaerobic metabolism. Among the three acids tested, formic acid exerted the most substantial inhibitory effect on *S. pneumoniae*. Nevertheless, the presence of serum enabled the bacteria to survive even at formic acid concentrations that are otherwise lethal.

### 3.3. Formic Acid Promotes the Intake of SAA1 by S. pneumoniae

To explore the mechanism behind SAA1 reduction, we examined the effect of formic acid, a major product of anaerobic metabolism, on SAA1 levels. Two laboratory strains, D39 (serotype 2) and TIGR4 (serotype 4), were cultured in THY medium containing 20% mouse serum and 0.05% formic acid. Both strains significantly reduced SAA1 concentrations after 12 h, with TIGR4 showing a more pronounced effect (Figure 3A). Given that *S. pneumoniae* is known to degrade IgA via proteases, we hypothesized a similar mechanism for SAA1 degradation [35]. However, adding a protease inhibitor, even at five times the standard concentration (1× working concentration), did not prevent the reduction in SAA1 in co-culture with *S. pneumoniae* (Figure 3B). This suggested that proteolysis might not be the primary mechanism.

To determine whether *S. pneumoniae* internalizes SAA1, we incubated the bacteria with AF488-labeled recombinant SAA1 (AF488-rSAA1). We measured the AF488 fluorescence intensity in both the supernatant and bacteria after a 3 h incubation with *S. pneumoniae* (Figure 4A). The results indicated a significant decrease in AF488 fluorescence in the supernatant following co-culture with TIGR4 (Figure 4B). Moreover, the fluorescence intensity of AF488-rSAA1 within the bacteria increased significantly after treatment with 0.05% Triton X-100, suggesting intracellular localization of AF488-rSAA1 (Figure 4C,D). After 1 h, AF488-rSAA1 was distributed around the bacteria; by 2 h, it adhered to the bacterial membrane. At 3 h, AF488-rSAA1 fully overlapped with the bacteria (Figure 4E). The picture at 3 h is distinct from the membrane adhesion observed at 2 h, with increased fluorescence intensity indicating transparency (Figure 4D). Taken together, it is inferred that *S. pneumoniae* begins ingesting AF488-rSAA1 after 3 h. However, it is important to acknowledge that only a small proportion of *S. pneumoniae* cells internalized SAA1 under the current experimental conditions. This may be due to limitations of the in vitro culture system, as more than 60% of the bacteria died even after 3 h of incubation in PBS without formic acid. (Appendix A) Therefore, the current setup may not accurately reflect the physiological environment, and further optimization of the culture conditions will be necessary to better investigate the interaction between *S. pneumoniae* and SAA1.

The findings indicate that formic acid enhances *S. pneumoniae*’s capacity to internalize SAA1. Notably, the TIGR4 strain exhibits a greater ability to lower SAA1 levels compared to D39, prompting its selection for investigating the underlying mechanism. Using ELISA to measure SAA1 fold changes post-co-culture (Figure 3A,B) and fluorescence microscopy-localized AF488-rSAA1, as well as to assess bacterial interaction, it is evident that *S. pneumoniae* decreases SAA1 concentrations in BALF through intaking SAA1.

### 3.4. Recombinant SAA1 Enhances S. pneumoniae Tolerance to Formic Acid

We next investigated whether intake of SAA1 contributes to the enhanced acid resistance of *S. pneumoniae* observed in the presence of serum. Using recombinant SAA1, we tested its ability to protect *S. pneumoniae* from formic acid stress in PBS containing 0.05% formic acid. Colony-forming units (CFUs) were measured at 1 and 3 h. The addition of 25 μg/mL recombinant SAA1 significantly improved bacterial survival at both time points (Figure 5A and Appendix A). To visualize this interaction, we stained dead bacteria with Propidium Iodide and incubated the bacteria with 25 μg/mL AF488-rSAA1 in PBS containing a lethal concentration of 0.075% formic acid. Fluorescence microscopy showed that bacteria containing internalized AF488-rSAA1 were more resistant to the lethal acidic environment (Figure 5B). Interestingly, we observed that SAA1 concentrations higher than 25 µg/mL exhibited a detrimental effect on *S. pneumoniae*, consistent with previous reports describing the bacteriostatic properties of SAA1 [31,32]. These findings suggest that while sub-physiological levels of SAA1 (around 25 µg/mL, Appendix A) may support bacterial growth or survival, supra-physiological concentrations appear to be harmful to the bacteria. Quantitative analysis revealed that approximately 20% of *S. pneumoniae* remained viable after 3 h of incubation in PBS containing a lethal concentration of 0.075% (*v*/*v*) formic acid. Among the surviving population, around 60% had internalized SAA1. Notably, all SAA1-internalized *S. pneumoniae* survived, indicating that SAA1 internalization may be essential for survival under acidic stress (Figure 5C).

These findings indicate that internalized SAA1 protects *S. pneumoniae* from acidic stress, particularly from formic acid. This mechanism likely underlies the serum-mediated enhancement of bacterial survival under anaerobic conditions.

## 4. Discussion

In this study, we revealed a previously uncharacterized mechanism by which *S. pneumoniae* utilizes host acute-phase proteins to adapt to acidic environments. Specifically, our data showed that *S. pneumoniae* can internalize SAA1, thereby enhancing its resistance to formic acid stress, some byproducts of its anaerobic metabolism. This finding highlights a potential survival strategy whereby the pathogen utilizes host immune components for its own benefit.

SAA1 is an acute-phase protein involved in innate immunity. It activates monocytes and macrophages through receptors such as TLR2, TLR4, and formyl peptide receptor 2 (FPR2), promoting NF-κB signaling and driving local inflammation. In neutrophils, SAA1 induces chemotaxis, degranulation, and reactive oxygen species (ROS) release via FPR2 and it also facilitates dendritic cell maturation, enhancing antigen presentation through the upregulation of MHC II and co-stimulatory molecules like CD80/CD86. Additionally, SAA1 plays a critical role in T cell polarization, particularly promoting Th17 differentiation [36,37,38,39,40,41,42,43,44,45,46].

Our findings indicated that *S. pneumoniae* can internalize SAA1, especially in acidic environments enriched with formic acid. Previous studies reported that SAA1 binds to the outer membrane protein A (OmpA) of Gram-negative bacteria, such as *Escherichia coli* and *Pseudomonas aeruginosa*, enhancing neutrophil-mediated phagocytosis [31]. Interestingly, under acidic conditions, SAA1 also binds the membrane of *Staphylococcus aureus* and inhibits its growth [32]. In mice, the physiological concentration of SAA1 ranges from approximately 0.2 to 0.8 µg/mL in bronchoalveolar lavage fluid (BALF) and from 5 to 25 µg/mL in serum (Figure 1C and Appendix A). In this study, we supplemented THY broth with 20% mouse serum to mimic the SAA1 concentration found in BALF. This approach allowed us to investigate the potential impact of physiologically relevant levels of SAA1 on *S. pneumoniae* growth and survival under controlled conditions. In our model, co-culturing *S. pneumoniae* with recombinant SAA1 in the presence of formic acid led to a significant decrease in extracellular SAA1 and a concomitant increase in intracellular fluorescent signal, suggesting internalization. Our data also demonstrated that recombinant SAA1 enhances *S. pneumoniae* survival under formic acid stress. This protective effect is observable both at the population level (via CFU enumeration) and at the single-cell level using fluorescently labeled SAA1. Interestingly, we found that bacterial resistance to formic acid stress was most enhanced at SAA1 concentrations close to those found in serum. This observation raises the possibility that *S. pneumoniae* may exploit SAA1-mediated mechanisms to resist acidic stress after entering the bloodstream. These results imply a functional advantage conferred by SAA1 intake, potentially allowing the pathogen to counteract environmental acidity and evade immune activation.

However, several limitations warrant consideration. First, we did not elucidate the molecular pathway through which *S. pneumoniae* internalizes SAA1. Murine SAA1 is a small protein consisting of 105 amino acids with a molecular weight of approximately 12,028 Da, suggesting the possibility that it may be taken up via a permease transport system. A plausible candidate is the Ami-AliA/AliB permease system, known for peptide transport in *S. pneumoniae* [47]. Future work will involve generating deletion mutants of this system to assess its role in SAA1 uptake. Furthermore, it remains unclear whether SAA1 must be cleaved before uptake or whether intact protein is sufficient. Although fluorescence microscopy provided initial evidence of intracellular localization, resolution limitations prevented detailed spatial analysis. Techniques such as bacterial cell expansion may allow more precise visualization in future studies. Additionally, SAA1 levels showed variability among individual mice, which we could not fully explain. Nonetheless, across three independent experiments with six mice per group, the trend of reduced BALF SAA1 following *S. pneumoniae* infection remained consistent.

This study pioneers the exploration of *S. pneumoniae* interactions with SAA1 and introduces the concept that pathogens may exploit host immune proteins to survive hostile environments. Our findings advance the understanding of the symbiotic relationship between *S. pneumoniae* and its host. By ingesting SAA1, *S. pneumoniae* prevents the activation of other immune cells that could target it, thereby resisting formic acid stress. This mechanism might also be present in other pathogenic bacteria, particularly intestinal pathogens. Some bacteria may similarly ingest proteins to generate ammonia, neutralizing acidic environments in the intestine. Bacteria primarily utilize urease to withstand acidic environments. *Helicobacter pylori*, for instance, survives gastric acid by producing urease, which decomposes urea into ammonia (NH) and carbon dioxide (CO_2_), with ammonia neutralizing stomach acid [48]. In the colon, unabsorbed proteins and peptides are fermented by intestinal flora, producing ammonia that elevates local pH to combat acid stress [49].

In conclusion, this study provides new insights into the adaptive strategies of *S. pneumoniae*, suggesting that host-derived SAA1 may serve a dual role, both as an immune signal and as a bacterial survival factor. Elucidating these interactions may open new avenues for antimicrobial strategies that target pathogen–host protein interactions and metabolic adaptations.

## Figures and Tables

**Figure 1 microorganisms-13-01309-f001:**
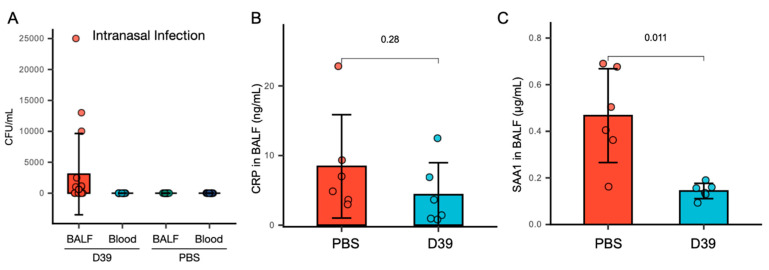
SAA1 decreased but CRP was not significantly changed in 600 µL BALF collected from intranasal *S. pneumoniae* infection mice. (**A**) Detection of *S. pneumoniae* in both blood and BALF 24 h post-PBS- or *S. pneumoniae* D39-nasal administration. n = 18. All individual data points are non-negative. Data are presented as mean ± SEM. (**B**) CRP in BALF displayed no significant difference between non-infection (PBS-intranasal administration, n = 6) and infection (*S. pneumoniae* D39-nasal administration, n = 6) groups. Student’s *t*-test was used for statistical analysis (*p* = 0.28, no significant difference). The experiment was independently repeated three times with similar trends observed. The data presented are from one representative experiment. (**C**) SAA1 in BALF decreased in infection (*S. pneumoniae* D39-nasal administration, n = 6) group in comparison to non-infection (PBS-intranasal administration, n = 6) group. Student’s *t*-test was used for statistical analysis (*p* < 0.05, showed significant difference). The experiment was independently repeated three times with similar trends observed. The data presented are from one representative experiment.

**Figure 2 microorganisms-13-01309-f002:**
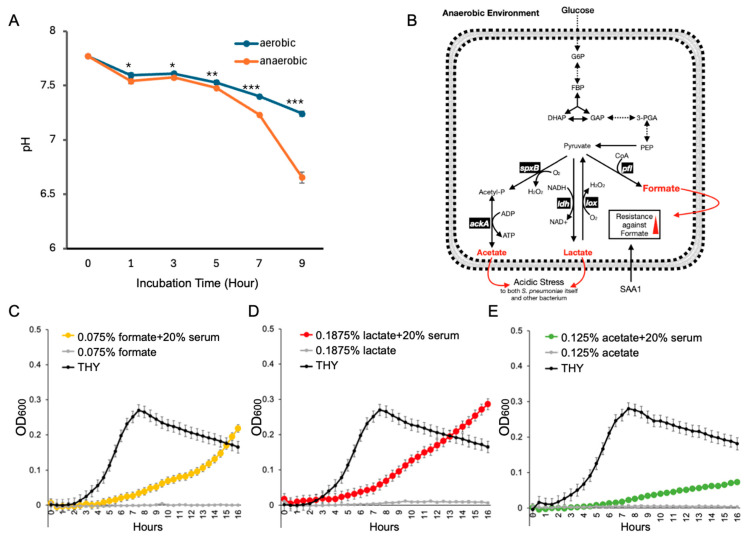
Serum components facilitate *S. pneumoniae* resistance against acidic stress. (**A**) Bacterial cultures under anaerobic conditions showed more acidity in comparison to that in anaerobic conditions. Anaerobic conditions were created using an anaerobic jar equipped with a Anaero Pack (Mitsubishi Gas Chemical Co., Inc., Tokyo, Japan). n = 6. *, *p* < 0.05; **, *p* < 0.01; ***, *p* < 0.005; Student’s *t*-test was used for statistical analysis. (**B**) Graphic summary of *S. pneumoniae* anaerobic metabolism described in KEGG database. (**C**) Growth curve of *S. pneumoniae* in lethal concentrations of formate with/without serum components. (**D**) Growth curve of *S. pneumoniae* in lethal concentrations of lactate with/without serum components. (**E**) Growth curve of *S. pneumoniae* in lethal concentrations of acetate with/without serum components.

**Figure 3 microorganisms-13-01309-f003:**
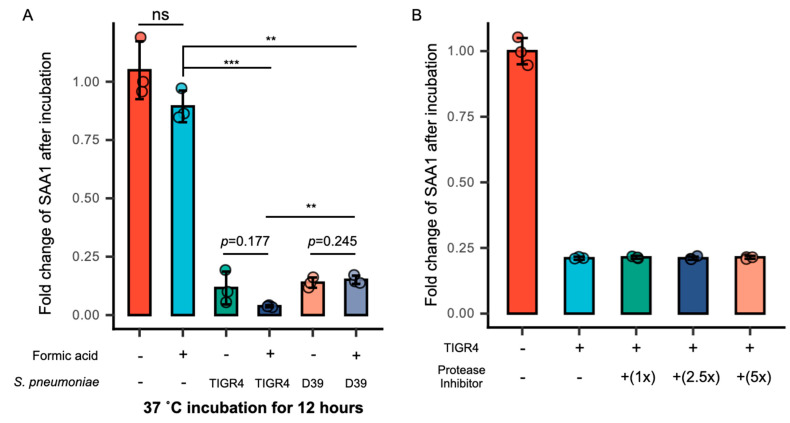
The reduction in SAA1 level is independent of protease. (**A**) TIGR4 strain displayed stronger capacity to cause SAA1 reduction with the presence of formic acid (0.05%, *v*/*v*) in vitro compared to D39 strain. n = 3; ns, no significant difference; **, *p* < 0.01; ***, *p* < 0.005; Student’s *t*-test was used for statistical analysis. (**B**) SAA1 reduction was not affected by the addition of a protease inhibitor in the presence of 0.05% formic acid, which indicates SAA1 reduction was not mediated by bacterial protease. n = 3. Student’s *t*-test was used for statistical analysis.

**Figure 4 microorganisms-13-01309-f004:**
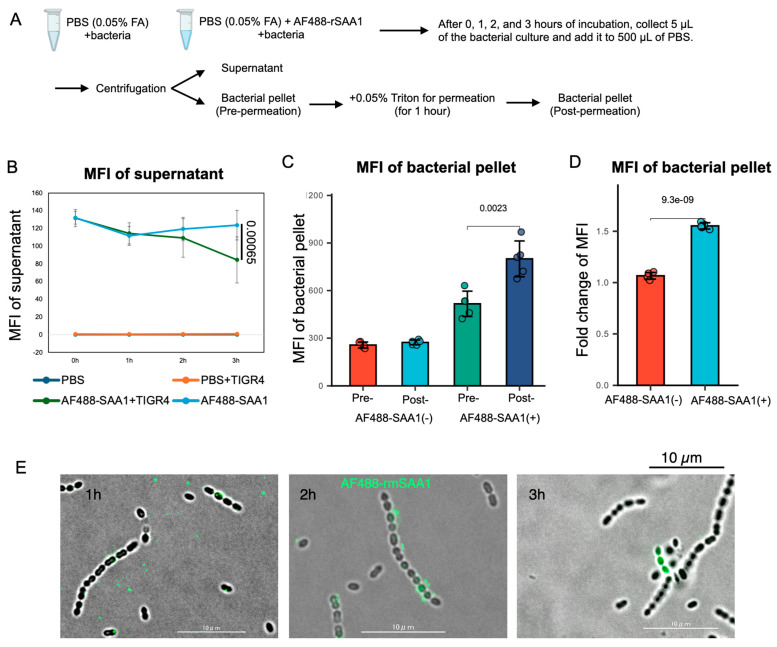
*S. pneumoniae* reduces SAA1 levels via intake rather than catalysis. (**A**) Experimental workflow. (**B**) Mean Fluorescence Intensity (MFI, Ex/Em = 485/535) of supernatant collected from co-culture system containing *S. pneumoniae* and AF488-rSAA1. At 3 h after incubation at 37 °C, the MFI of AF488-rSAA1 co-cultured with bacteria showed a significant decrease in comparison to the control group without bacteria. (**C**) MFI of bacterial cell pre- and post-permeation using 0.05% Triton. (**D**) Change in MFI (Ex/Em = 485/535) of the bacterial cell after permeation using 0.05% Triton. (**E**) Fluorescence microscope showed the location of AF488-rSAA1 at 1, 2, and 3 h post-incubation at 37 °C.

**Figure 5 microorganisms-13-01309-f005:**
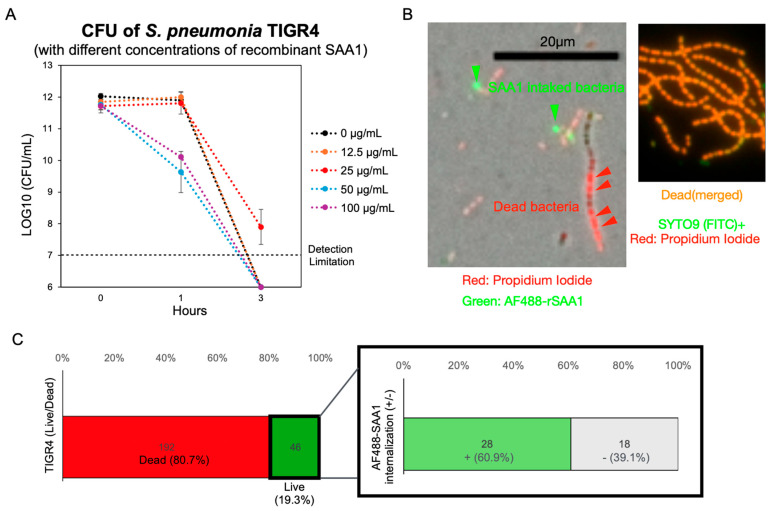
Recombinant SAA1 enhance *S. pneumoniae* resistance against formic acid. (**A**) CFU counting of *S. pneumoniae* TIGR4 co-cultured in PBS containing 0.05% formic acid and different concentrations of recombinant SAA1. The CFUs of each group were recorded at 0 h, 1 h, and 3 h. n = 3. (**B**) A fluorescence microscope showed that the intake of AF488-rmSAA1 keeps *S. pneumoniae* alive in a lethal concentration of 0.075% formic acid. The left image showed the experimental group (*S. pneumoniae* was incubated with AF488-rmSAA1) and the right image showed dead bacteria as a control (*S. pneumoniae* was incubated with 70% ethanol for 15 min at room temperature). Green, AF488-rSAA1; red, dead bacteria; orange, merging of red and green fluorescence. (**C**) Survival of *S. pneumoniae* TIGR4 in PBS containing 0.075% (*v*/*v*) formic acid and 25 µg/mL SAA1 after 3 h of incubation at 37 °C. The left panel shows the percentage of live (unstained) and dead (Propidium Iodide-stained, red) bacteria. The right panel displays the proportion of SAA1-positive (green) and SAA1-negative (unstained) cells among the live bacterial population. Notably, no bacteria were simultaneously stained with both red and green signals, indicating that all SAA1-positive cells survived under these conditions.

## Data Availability

The original contributions presented in this study are included in the article/Appendix A. Further inquiries can be directed to the corresponding author.

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
