# Peer review of "Host Serum Amyloid A1 Facilitates Streptococcus pneumoniae Adaptation to Acidic Stress Induced by Pneumococcal Anaerobic Metabolism"

_microorganisms, 2025, doi:10.3390/microorganisms13061309_

Round 1
Reviewer 1 Report
Comments and Suggestions for Authors
Comments to Gong et al.
The aim of this study was to investigate the role of the acute phase protein SAA1 during the early infection phase of Streptococcus pneumoniae. The authors claim that the SSA1 level in BALF is reduced upon S. pneumoniae infection, and this reduction is due to bacterial uptake of SAA1. They further claim that the SSA1 taken up by the bacteria protects the bacteria from acidic stress caused by formic acid. The latter is based on microscopic imaging. The latter claim needs to be better substantiated by quantification and flow cytometric analysis. The conclusions is overinterpreted and should be tuned down.
Other comments:
Line 18: I would use another word than "employed", e.g., "used".
Line 27: Please correct to: "is internalized" ("is" is lacking – passive tense).
Line 37: I am not sure if "briefly" is the correct word to use here. Please check the sentence.
Line 37: The following part of the sentence " If it breaches the upper respiratory defenses" needs more detailed description.
Line 41: The sentence needs a reference. Also, how the neutrophil function was impaired should be mentioned.
Line 45: There is an accumulation of fluid in the alveoli. You can't say "extravasate accumulation". Extravasate – is the flow of fluid from the vessel to the tissues.
Line 47: I think you can delete " which is a symptom in some bacterial pneumoniae patients" which is superfluous.
Line 53: It would be worthwhile if you could add some mechanisms how S. pneumoniae is tolerant to oxidative stress.
Line 55: Please correct to: "which is catalyzed by".
Line 60-63: The sentence is not logical. "Despite" should be followed by a contradiction, but there is no contradiction. It survives under acidic conditions and tolerates extreme pH. The mechanism for the acid tolerance response should be described.
Line 63: The "acidic-stress induced lysis" should be better described, especially how the "lysis" is compatible with the "survival capacity".
Sentence ending in line 69 should have a reference.
Line 78: I think you meant "likely" instead of "potentially".
Lines 81-82: Maybe you intended: "protection during the early infection phase".
Line 100: You need to state the initial OD of the culture. How did you distinguish between the low OD value of 0.1 during the lag phase and the early exponential phase? For how long did you incubate them after dilution?
Line 105 has a spelling mistake. It is written: " then mice were given to mice". I think you meant " then bacteria were given to mice"
Line 109: You need to mention how much PBS was injected. If you have injected PBS, then the BALF is diluted. You need to describe how you calculated the dilution factor.
Line 110: You need to state how you collected the blood.
Line 110: Please correct to: "Serum was obtained"
Line 119: Please correct to: "Serum was collected".
The title of Section 2.4 does not conform with the text. Please correct. It is not a "co-culture" but an incubation in THY broth containing 20% serum.
Line 124: Why was formic acid added here? What was the control condition?
Line 126: The concentration of the protease inhibitor cocktail should be stated.
Line 128: Correct to "supernatant" and "was".
Line 129: Please correct to: "was".
Title 2.5: "observation" can be deleted.
Line 135: Please correct to "was" instead of "were".
Line 139: Instead of "with existence" I would suggest writing "in the presence of".
Line 139: I would write: "a non-toxic concentration". You also need to state the controls.
Line 141: space should be added before "incubator".
There are several syntactic errors in Section 2.5.
Line 142: Instead of "then were" I would suggest writing "and".
Line 143: Please avoid using "we". Also, the sentence mentions only 3 hours, while in previous sentence it says 1, 2 and 3 hours. Please correct. You only took 5 μl of bacteria, so did you really centrifuge them? Why did you need to permeabilize the bacteria? If the bacteria take up the fluorescent SAA1, there should not be a need for permeabilization. If you do permeabilization, it is a non-physiological process of mere diffusion.
Line 147: You need to rephrase in order not to include here the figure number. The figure number should appear in consequent order.
Line 153: "was" instead of "were". The volume and number of bacteria added should be stated.
Line 156: I think you intended "solution" instead of "broth". The solute of the phenol red solution should be stated.
Line 157: "was" instead of "were".
Line 158: Please correct to: "and the absorbance at 560 nm was measured".
Line 161: You need to show that there is a pH-dependent stain intensity of phenol red (the calibration curve).
Section 2.8 should include the number of replicates and the number of experimental repetitions.
Line 183: The bacterial name should not be in bold. The same for other titles.
Line 194: Instead of "remained stable" I would suggest writing "was not significantly changed".
Figure 1: The volume of BALF collected should be stated.
Line 107: "potentially" is not a proper word. Please rephrase the sentence.
Line 212: There is an extra point that should be deleted.
Line 213: The question is whether the environment is really anaerobic (total lack of oxygen), or there is only a reduction in the oxygen pressure. A reference should be added here.
Line 220: Does the serum neutralize the acids? What is the pH with or without serum? If serum neutralizes the acids, then it is not an "adaptation to acidic conditions".
Figure 2: You need to state how you made the "anaerobic" condition in these experiments.
Line 245: You need to state what the standard concentration is. Could it be due to the presence of autolysins or a hydrolase? A degradation of SAA1 can be analyzed by running a protein gel. You can incubate the SAA1 with the supernatant of the bacteria with or without formic acid, and then quantify the SAA1 level. This will provide a better answer to whether some secreted factors by the bacteria degrades SAA1.
Line 248: "independent of"
Line 249: "the presence of" instead of "with existence of".
Line 253: "was used"
Line 255: Please add space before the parenthesis.
Figure 3: You need to prove that the proteases secreted by the bacteria are neutralized by the protease inhibitor cocktail. How does formic acid affect protease secretion from the bacteria?
Line 259: If the bacteria can take up SSA1, why should you need to add 0.05% Triton X-100 which makes pores? Could it be that formic acid makes pores too? This needs to be stated in the text.
Figure 4. A. The text in Y-axis states "without bacteria". But you have added bacteria to 3 of the samples. You also need to show the histograms. The same for B. MFI should be defined. The legend should include how you measured the MFI. Why did you write twice Ex/Em=Ex/Em=? Isn't it sufficient to write once Ex/Em=? B. The true numbers of MFI should also be stated. Quantification by flow cytometry is needed. A change of 1.5 in MFI is not really a big change.
According to Figure 4C, only some bacteria absorb SSA1. The percentage of bacteria taking up SAA1 should be stated. This can be measured by flow cytometry. Can really this small percentage of SAA1 uptake be responsible for the reduced SAA1 level?
Figure 4: What is the viability of the bacteria in PBS? You need to state in the legend the incubation temperature.
Line 274: The authors claim that formic acid enhances S. pneumoniae's capacity to degrade SAA1. But, in a previous sentence they claim that proteases are not involved, but the bacteria take up SSA1. Please clarify.
Line 278: Spelling mistake. Correct to: Localized.
For how long is SAA1 intact in the bacteria. Is it degraded? Does it binds protons?
Line 287: Correct to: "lethal concentration of".
Figure 5: A. How can it be that 25, but not 50 or 100 μg/mL SSA1 protected the bacteria? 50 and 100 μg/mL SSA1 even increased bacterial death. The legend of B should be more precise. You need to state after how long the bacteria were exposed to SSA1 and the concentration of formic acid. Also, it is not clear the differences between the two images in B. This should be quantified. You need to show control images for comparison. Flow cytometry would provide quantitative data. Images for the exposure to formic acid should also be shown. Was permeabilization with Triton X-100 done here?
Line 308: Please spell out FPR2.
More relevant references should be added to discussion.
Line 314: Spelling mistake. Please correct to: "internalize".
The increase in SAA1 internalization by formic acid should be quantified and compared to bacteria under control conditions. Could it be due to membrane permeabilization.
Line 322: Is endocytosis a process that can be performed by bacteria. Please provide data that can support this. They have a cell wall that usually is impermeable, and there are several membranal transport systems.
Do you have a possibility to follow the fate of bacteria which have taken up SAA1?
Line 330-: You need to state the size of SAA1.
You need to state the pH of the different acid solutions used in the text.

There are some grammar and typo error. Except for this, the English in general is fine.
Author Response
We thank the reviewer for the careful evaluation of our manuscript and for the constructive comments regarding the interpretation and substantiation of our findings. We appreciate your acknowledgement of our aim to explore the role of the acute-phase protein SAA1 during the early phase of Streptococcus pneumoniae infection.
We acknowledge the reviewer’s concern regarding the interpretation of our data, particularly the conclusion that bacterial uptake of SAA1 confers protection against acidic stress caused by formic acid. In response, we have toned down the relevant statements in the manuscript to avoid overinterpretation and have clarified the limitations of our current evidence.
Please find our detailed responses and the corresponding changes to the manuscript below.

Reviewer 2 Report
Comments and Suggestions for Authors
The manuscript by Weichen Gong and co-authors titled "Host serum amyloid A1 facilitates Streptococcus pneumoniae adaptation to acidic stress induced by pneumococcal anaerobic metabolism" is devoted to the investigation of the role of serum amyloid A1 in immune modulation during pneumococcal infection in mice. The study is interesting and in some ways even pioneering, and I found no major issues with it. Below are some minor remarks and corrections, primarily concerning formatting and clarity:
Abstract: Remove the headings Background, Methods, Results, Conclusion.
First mention of Streptococcus: Write the full name (Streptococcus pneumoniae).
References: Use square brackets (e.g., [1]) instead of superscript.
Lines 80–84: Clarify the study’s aim more precisely.
Line 85: Avoid unconventional phrasing - simply title the section Materials and Methods.
Lines 94, 153: In subsequent mentions, abbreviate Streptococcus to S. (e.g., S. pneumoniae).
Line 94: Specify the source of the D39 and TIGR4 strains.
"Mice were given to mice": This phrasing is unclear. Revise for clarity.
Microscope model: Indicate the specific model used (e.g., BZ-X700, BZ-X710, BZ-X800, or BZ-X810).
Line 147: Figure labels should follow sequential numbering (e.g., Figure 1, 2, etc.). Why does it start with 5B?
Lines 175–179: This sentence is confusing. Rewrite and consider splitting it into two.
Line 314: Did you mean internalize?
Author Response
We sincerely appreciate the reviewer’s thoughtful and encouraging comments on our manuscript titled “Host serum amyloid A1 facilitates Streptococcus pneumoniae adaptation to acidic stress induced by pneumococcal anaerobic metabolism.” We are grateful for your recognition of the novelty and significance of our study.
In response to your valuable suggestions regarding formatting and clarity, we have carefully revised the manuscript to address all the minor issues you highlighted. Please find below our point-by-point responses and corresponding changes made to the text.

Round 2
Reviewer 1 Report
Comments and Suggestions for Authors
The manuscript has been improved and most of the comments have been addressed properly.
In the response letter, the authors claim that there are supplementary data, but I was unable to find them. For instance, the calibration curve for pH-dependent stain intensity of phenol red is claimed to appear in the supplementary data file.
How can the standard deviation in Figure 1A be below 0?
Figure 3A lacks the fold change of SAA1 after incubation with bacteria in the absence of formic acid.
Legend to Figure 3B should state if this experiment was done in the absence or presence of formic acid.
Line 303 in revised version: Still it is written “degrade” which is in contrast to the protease inhibitory cocktail study, showing that the main reason for the reduced SAA1 level is due to bacterial uptake. The text should be corrected. The figure presented in the response letter can be added to the Supplementary data.
The pH measurements presented in the response letter should be added to the supplementary data and be referred to in the text.
I think it is important to emphasize in the text that only a small fraction of the bacteria take up the fluorescent SAA1. A quantification of this ought to be done (Figure 4).
A quantification of Figure 5B should also be done. As I asked in previous report, more images would be appreciated.
Why is there a detection limitation in Figure 5A? A lower dilution should have been done.
The duality of SAA-1 on bacterial survival should be better emphasized in the result section.
Another point that should be discussed is the much higher concentrations of SAA1 used in Figure 5 in comparison to the SAA1 level in BALF (Figure 1).
Author Response
Thank you very much for taking the time to carefully review our revised manuscript. The reviewer’s constructive comments have indeed improved the rigor and readability of our manuscript. We sincerely apologize for not including the supplementary data in our previous submission. In response to the latest revision requests, we have marked the newly revised content in green (while the previous revisions are indicated in red) for clarity. We hope this will help the reviewer to easily track the changes.
